# Association of Serum BAFF Levels with Cardiovascular Events in ST-Segment Elevation Myocardial Infarction

**DOI:** 10.3390/jcm12041692

**Published:** 2023-02-20

**Authors:** Ziyang Wang, Yueying Wang, Yuke Cui, Zhiyong Chen, Lei Yi, Zhengbin Zhu, Jingwei Ni, Run Du, Xiaoqun Wang, Jinzhou Zhu, Fenghua Ding, Weiwei Quan, Ruiyan Zhang, Jian Hu, Xiaoxiang Yan

**Affiliations:** 1Department of Cardiovascular Medicine, Ruijin Hospital, Shanghai Jiao Tong University School of Medicine, Shanghai 200025, China; 2Institute of Cardiovascular Diseases, Shanghai Jiao Tong University School of Medicine, Shanghai 200025, China

**Keywords:** B cell activating factor, acute myocardial infarction, prognosis

## Abstract

Objectives: The B cell activating factor (BAFF) is a B cell survival factor involved in atherosclerosis and ischemia-reperfusion (IR) injury. This study sought to investigate whether BAFF is a potential predictor of poor outcomes in patients with ST-segment elevation myocardial infarction (STEMI). Methods: We prospectively enrolled 299 patients with STEMI, and serum levels of BAFF were measured. All subjects were followed for three years. The primary endpoint was major adverse cardiovascular events (MACEs), including cardiovascular death, nonfatal reinfarction, hospitalization for heart failure (HF), and stroke. Multivariable Cox proportional hazards models were constructed to analyze the predictive value of BAFF for MACEs. Results: In multivariate analysis, BAFF was independently associated with risk of MACEs (adjusted HR 1.525, 95% CI 1.085–2.145; *p* = 0.015) and cardiovascular death (adjusted hazard ratio [HR] 3.632, 95% confidence interval [CI] 1.132–11.650, *p* = 0.030) after adjustment for traditional risk factors. Kaplan-Meier survival curves demonstrated that patients with BAFF levels above the cut-off value (1.46 ng/mL) were more likely to have MACEs (log-rank *p* < 0.0001) and cardiovascular death (log-rank *p* < 0.0001). In subgroup analysis, the impact of high BAFF on MACEs development was stronger in patients without dyslipidemia. Furthermore, the C-statistic and Integrated Discrimination Improvement (IDI) values for MACEs were improved with BAFF as an independent risk factor or when combined with cardiac troponin I. Conclusions: This study suggests that higher BAFF levels in the acute phase are an independent predictor of the incidence of MACEs in patients with STEMI.

## 1. Introduction

Over the past few decades, advances in pharmacologic, catheter-based, and surgical reperfusion have improved outcomes for patients with acute myocardial infarctions (AMI). Despite all the improved treatments, the prognosis of patients with AMI is still poorer than that of most other diseases. Complications of AMI, such as heart failure (HF) and cardiovascular events, could further affect the quality of life for patients. At present, the ability to identify risk-prone patients remains limited, and further exploration of novel biomarkers may help identify these high-risk patients.

The immune response and inflammatory process in the post-AMI period are regulated by different classes of immune cells, cytokines, and chemokines. Mature B lymphocytes alter the recovery of cardiac function after AMI [1]. The proliferation and activation status of B cells are key factors in determining cardiovascular disease (CVD) risk [2]. B-cell-activating factor (BAFF, also known as B lymphocyte stimulator-Blys) is a member of the tumor necrosis factor (TNF) superfamily, crucial for the survival, proliferation, and differentiation of B cells [3,4]. Its receptors are critically involved in regulating B cell maturation and homeostasis [5,6]. Therefore, we speculated BAFF might have an effect on the prognosis of patients with AMI. BAFF has been previously studied in patients with autoimmune diseases, including rheumatoid arthritis and lupus [7,8], which exhibit a strong predisposition to premature atherosclerosis and heart attacks [4]. Studies in hyperlipidemic atherosclerotic mice indicate a significant role for BAFF in the development of atherosclerotic lesions, in particular, vulnerable lesions that are prone to rupture, causing heart attacks [9]. Here, we addressed the role of BAFF, which orchestrates a variety of adaptive immune responses relevant to human diseases but that has relatively been neglected in the setting of ischemic injury. Thus, the present cohort study aimed to examine the relationship between circulating BAFF and adverse outcomes in the post-MI period.

## 2. Methods

### 2.1. Study Population

This prospective cohort study consecutively enrolled eligible patients who were admitted because of STEMI to the Critical Cardiac Care Unit and underwent coronary artery angiography (CAG) at the Ruijin hospital affiliated with Shanghai Jiao Tong University School of Medicine (Shanghai, China) between July 2016 and June 2017. The diagnoses of STEMI were made according to the Third Universal Definition of Myocardial Infarction [10]. Moreover, all patients enrolled were 18 years of age or older. Exclusion criteria included patients with a severe physical disability, other serious diseases such as malignant tumors and autoimmune diseases, or who were unwilling to participate were excluded. The research was approved by the Institutional Review Board of Ruijin Hospital, Shanghai Jiao Tong University School of Medicine. Each subject provided written, informed consent before enrolment.

### 2.2. Data Collection

All demographic and clinical characteristics were collected through face-to-face interviews at admission, which was set as the baseline. Physical examinations were performed on the same day. Echocardiography and regular laboratory tests were performed on the 2nd day after admission. Each patient received standard care during STEMI hospitalization according to current guidelines [11,12]. The following data were collected: serum or plasma levels of glucose and lipids, liver and renal function, electrolytes, and cardiac troponin I (cTnI). The estimated glomerular filtration rate (eGFR) was calculated using the Chronic Kidney Disease Epidemiology Collaboration equation [13]. Body mass index (BMI) (kg/m^2^) was calculated using height and weight measured at baseline. The Syntax score was used to quantify the degree of each vascular lesion via coronary angiography.

### 2.3. Study Endpoints and Follow-Up

All patients were followed up for three years and scheduled at 1 month, 3 months, 6 months, 12 months, and every 6 months thereafter until the event occurred or to the last visits. Clinical events were collected via clinic visits, medical records, or telephone calls by research staff in order to record mortality, HF hospitalizations, and other adverse events. Moreover, the date of the event was recorded, and information explaining the cause was obtained.

The primary endpoint was major adverse cardiovascular events (MACEs), defined as a composite of cardiovascular death, nonfatal reinfarction, hospitalization for HF, and stroke. All endpoints were defined according to the proposed definitions by the Standardized Data Collection for Cardiovascular Trials Initiative (Appendix A) [14]. All events were independently evaluated by adjudicators blinded to the results of the BAFF levels. The adjudicators also reviewed the source documents and established the necessity for hospital admission.

### 2.4. BAFF Measurement

Blood samples were collected from the peripheral artery before CAG. After the blood culture sampling is completed, it should be sent to the biological laboratory for verification by a dedicated person immediately. Whole blood was left standing at room temperature for not more than 2 h, followed by centrifugation at room temperature at 2000 RPM for 15 min to obtain serum. The sera obtained by centrifugation were immediately stored at −80 °C for subsequent analysis. Serum BAFF concentration was measured using a human BAFF ELISA kit (Cat # CSB-E11912h) from CUSABIO. All samples were anonymized before being handled by technicians, who were blinded to the clinical characteristics of patients.

### 2.5. Statistical Analysis

The baseline characteristics of the entire cohort were summarized using standard descriptive statistics. We used Shapiro–Wilks to test the normality of continuous variables. Continuous variables were expressed as mean ± standard deviation (SD) if data were normally distributed, otherwise as the median and interquartile range (IQR). Categorical data were presented as counts and percentages. One-way analysis of covariance (ANOVA) or the Kruskal–Wallis test was used to compare continuous variables, as appropriate. The chi-square test or Fisher’s exact test was designed to analyze the differences in categorical variables. The BAFF levels were log-transformed or divided into tertiles for further analysis. Receiver operating characteristic (ROC) curves were obtained, and the associated cut-off points with the greatest sensitivity and specificity were selected according to Youden’s index. The area under the ROC curve (AUC) was calculated to determine the discriminative ability of BAFF in major adverse cardiovascular events.

Kaplan–Meier events-free survival analysis was performed using the log-rank test. The association between baseline BAFF level and the risk of MACEs was analyzed using the Cox proportional hazards model, with hazard ratios (HR) with 95% confidence intervals (CI) as the summary statistics. We used unadjusted, partially adjusted, and fully adjusted Cox proportional hazards models. These included: (1) age and gender (female [referent], male); (2) age, gender (female [referent], male), body mass index (BMI), smoking (no [referent], yes), hypertension (no [referent], yes), diabetes mellitus (no [referent], yes), hyperlipidemia (no [referent], yes), white blood cells (WBC), high sensitivity C-reactive protein (hs-CRP) and left ventricular ejection fraction (LVEF). We also explored the nonlinear dose-response relationship between the BAFF and the risk of MACEs using a restricted cubic spline (RCS) model with four knots (at the 5th, 25th, 75th, and 95th percentiles). The additional predictive new predictor over a reference model was assessed using Harrell’s C-statistics calculated from a Cox proportional hazards regression model [15]. The *p*-value of the C-statistics and IDI compared with the reference model was performed by a likelihood ratio test used for the Cox models [16].

A priori subgroup analyses were performed for age (<65 vs. ≥65), gender (male vs. female), diabetes (yes vs. no), hypertension (yes vs. no), and dyslipidemia (yes vs. no) for this STEMI cohort.

All statistical tests were two-tailed, and a *p*-value < 0.05 was considered significant. All statistical analyses were conducted with SPSS 22.0 (IBM, Armonk, NY, USA) and R language software (version 4.1.1).

## 3. Results

### 3.1. Baseline Characteristics

A total of 347 patients with AMI were recruited, with 16 (4.61%) lost to follow-up. Finally, a total of 299 STEMI patients were enrolled in the final analysis (Figure 1). All patients were separated into three groups according to the tertiles of the BAFF levels: tertile 1, ≤0.79 ng/mL; tertile 2, 0.79 < BAFF ≤ 1.38 ng/mL; and tertile 3, >1.38 ng/mL. The baseline characteristics are presented in Table 1, including the clinical and laboratory characteristics. The mean patient age was 65.43 ± 12.57 years. Of these, 237 (79.3%) were men. Subjects with the highest tertile of BAFF were older, with relatively higher heart rates, higher prevalence of hypertension, lower BMI, poorer renal function, more serious inflammation, and worse left ventricular function. Moreover, hs-CRP was higher in the highest tertile of BAFF. Cardiac biomarkers, including NT-proBNP, cTnI, and CK-MB, were elevated in tertile 3.

### 3.2. Association between Serum BAFF Levels and the Risk of MACEs

Figure 2 shows the ROC curve generated using BAFF to discriminate adverse outcomes after AMI. BAFF had a greater area under the ROC curve, and a cut-off value of 1.462 had a sensitivity of 78.3% and specificity of 52.2% for MACEs after AMI.

Table 2 summarizes the results of the Cox regression models evaluating the association between BAFF measures and the risk of MACEs. Both univariate and multivariable Cox proportional hazard models further indicated the prognostic value of BAFF as a continuous variable when divided into tertiles. Higher BAFF levels were independently associated with the risk for MACEs (adjusted HR 1.525, 95% CI 1.085–2.145; *p* = 0.015), as a continuous log-transformed variable, adjusted for the full model including age, sex, BMI, LVEF, and various conventional influencing factors. Complete information about the effect of all relevant factors is shown in Appendix A. When the BAFF level was divided into tertiles, there was a four-fold increased risk (adjusted HR 4.116, 95% CI 1.690–10.022; *p* = 0.002) in the fully adjusted model for MACEs in tertile 3 compared to tertile 1. Additionally, higher BAFF levels were also significantly associated with the risk for cardiovascular death (adjusted HR 3.632, 95%CI 1.132–11.650, *p* = 0.030) when the BAFF level was divided into cut-off values (1.46 ng/mL). When fully adjusted, the prognostic value was significant when BAFF was analyzed as a log-transformed continuous variable (HR 2.083, 95%CI 1.014–4.279; *p* = 0.046). However, the correlation was not held when BAFF was divided into tertiles.

Further analyses using restricted cubic splines in Cox regression revealed a consistent increase in MACEs (*p_for non-linearity_* < 0.0001, *p_overall_* < 0.0001; Figure 3A) and cardiovascular death (*p_for non-linearity_* = 0.0329, *p_overall_* = 0.0134; Figure 3B) at high log^2^ BAFF levels. However, the BAFF levels were not significantly associated with hospitalization for HF, nonfatal reinfarction, and stroke (*p_overall_* > 0.05; Appendix A).

Moreover, Kaplan-Meier survival analysis showed that the higher BAFF levels were significantly associated with an increased risk for MACEs (*p* < 0.0001; Figure 4A) and cardiovascular death (*p* < 0.0001; Figure 4C). The result was consistent with another Kaplan–Meier survival analysis stratifying by the cut-off value of BAFF (*p* < 0.0001; Figure 4B,D). However, the higher tertiles of BAFF levels were not significantly associated with hospitalization for HF, nonfatal reinfarction, and stroke (*p* > 0.05; Appendix A). If stratifying by the cut-off value of BAFF, the result was consistent (*p* > 0.05; Appendix A).

Furthermore, when combined with cTnI, BAFF provided improvement to predict the MACEs in the C-statistic analysis. Similar results were also found in IDI values (Table 3).

### 3.3. Subgroup Analyses

We performed additional subgroup analyses according to age, gender, hypertension, diabetes mellitus, and dyslipidemia. Subgroup analyses demonstrated that the association of BAFF levels with MACEs was stronger among those without dyslipidemia (HR 1.639, 95% CI 1.292–2.078, *p_interaction_* = 0.029). There was no interaction by age, sex, diabetes, and hypertension on the association of BAFF levels with MACEs (Figure 5).

## 4. Discussion

Our present study, which enrolled patients with STEMI, demonstrated for the first time that BAFF could augment conventional risk stratification models in patients with STEMI, which may further contribute to the participation of BAFF in human myocardial infarction and its progression. During a three-year follow-up after AMI, higher BAFF levels were independently associated with the risk for MACEs and cardiovascular death, which enhanced current prediction models. Importantly, the consequences remained statistically significant after fully adjusting for traditional demographic factors and laboratory and imagological examination, including age, sex, history of hypertension and diabetes, and LVEF.

We chose to examine BAFF, a TNF family molecule of predominantly myeloid origin. BAFF and its receptors are critically involved in regulating B cell maturation and homeostasis [5,6]. Animal studies have revealed that overexpression of BAFF results in multiple autoimmune disorders [17,18]. In addition, Li et al. [19] showed an unexpected role of the BAFF signal in the central nervous system during ischemia injury. Under ischemic brain conditions, the BAFF was vastly upregulated in microglia, and this upregulation could at least be attributed to the JAK-STAT signaling pathway activated by IFN-c and IL-10 [19]. B-cell-targeted therapies are used in treating autoimmune diseases as well as lymphoid cancers [18,20,21,22], and some scholars have put forward the opinion that B-cell-targeted therapies might have potential applications in treating cardiovascular diseases [23].

In the atherosclerosis/lupus-prone mouse model, researchers demonstrated that the BAFF-BAFFR (B-cell–activating factor receptor) signal in B cells and the BAFF-TACI signal in macrophages play a double-edged role in both atheroprotective and proatherogenic effects [24]. Furthermore, high BAFF serum levels, as well as BAFF genetic variants, were found to increase susceptibility for systemic lupus erythematosus subclinical atherosclerosis even if all disease-related confounding factors were considered [4]. BAFF is targeted in the clinic for the treatment of systemic lupus erythematosus [25]. In a cardiovascular context, there is a large body of evidence demonstrating the involvement of lymphocytes in the pathogenesis of atherosclerosis; inflammatory lymphocytes accelerate the development and progression of atherosclerosis as well as contribute to lesion inflammation, and vulnerable plaque development, both B and T cells are involved [26,27]. B2 B cells promote the development/progression of atherosclerosis in hyperlipidemic mice by secreting inflammatory cytokines such as TNF-α and pathogenic antibodies [9,28]. Application of anti-B-cell-activating factor receptor (BAFF-R) antibody reduces the lesions 29, and depletion of B cells reduces the progression of atherosclerosis in mice [26]. Previous research on rodent models demonstrated that although the deletion of BAFFR attenuated the progression of atherosclerosis [29], neutralizing BAFF with an anti-BAFF antibody induced advanced atherosclerosis in both Apoe^−/−^ and Ldlr^−/−/^ mice. This indicates that soluble BAFF can influence atherosclerosis and cardiovascular health and regulate B cell immunity [30]. Natorska et al. speculate that B cells, which are in abundance in perivascular adipose tissue [31], and BAFF-expressing macrophage accumulation contribute to increased inflammation and thrombosis, leading to vessel calcification and further progression of the disease [32]. In addition, patients with low coronary flow reserve values express higher amounts of BAFF in plasma microvesicles [33]. In recent years, the incidence of AMI has been on the rise, with very high mortality and disability rates, and has become one of the main causes of HF [34]. In a model of ischemia-reperfusion (IR), the level of myocardial B cells increased and peaked between days 3 and 5 after injury and soon returned to baseline [35]. Furthermore, acute myocardial injury induced activation of the BCR signal in CD^11b−/^ cells and the TLR signaling pathway in CD^11b+^ cells [36]. This suggests that the subgroups of B cells in the heart after injury might have different functions. The current research indicates that activated B cells participate in the sustained state of myocardial inflammation and immune system activation after AMI and may affect the metabolism of myocardial collagen after AMI by secreting cytokines [37]. Moreover, B cells promote the expression of myocardial collagen Type I and Type III and damage the left ventricular ejection function [37]. However, the mechanism of action of B cells and their cytokines in the myocardial infarction process after AMI is still unclear. Moreover, it is unclear if BAFF affects the longer-term outcomes of AMI.

After being fully adjusted for several common risk factors in the Cox regression model, our research found that the elevation of BAFF was not only associated with MACEs but also cardiovascular death (Log^2^ BAFF: adjusted HR 2.083, 95%CI 1.014–4.279, *p* = 0.046). Compared with BAFF < 1.46 ng/mL, BAFF ≥ 1.46 ng/mL was associated with MACEs (adjusted HR 2.705, 95%CI 1.518–4.822, *p* = 0.001) and cardiovascular death (adjusted HR 3.632, 95%CI 1.132–11.650, *p* = 0.030). This means that patients with higher BAFF levels at admission had a higher risk of adverse outcomes. However, the highest tertile BAFF was not associated with cardiovascular death might cause by a bias that was attributed to the number of adverse outcomes in each group being quite small. Furthermore, when combined with other biomarkers, such as cTnI and NT-pro-BNP, the predictive power for MACEs has improved. This suggests that conventional biomarkers do not fully meet the requirement for risk stratification [38].

BAFF has been proven to play an important role in the process of atherosclerotic lesions. The patients included in this study may have plaque load or different degrees of coronary artery stenosis before admission, which foreshadows myocardial infarction. At the same time, this study found that the level of BAFF in patients with AMI was higher. Based on the close relationship between BAFF and atherosclerosis, BAFF may also have the value of early identification of patients with a high risk of myocardial infarction. The occurrence of cardiovascular disease is a continuous pathophysiological change. BAFF may not only predict the long-term prognosis after myocardial infarction but also identify high-risk coronary artery disease patients. The discovery of such early disease load biomarkers will help to push disease management to the preclinical stage, early detection and early intervention, early detection, and early treatment.

In addition, the strongest predictors of long-term survival in STEMI could be a consideration. In the consequent years, the number of FFR and IVUS procedures performed in patients with ACS increased. Performing FFR and IVUS in ACS does not significantly affect 30-day or one-year mortality [39]. High galectin-3 levels were associated with the short-term and long-term development of adverse cardiovascular events, heart failure, and re-hospitalization [40].

In all, the above results suggest that BAFF might be involved in the post-MI process by influencing immune response and inflammation, and cardiac remodeling, leading to cardiovascular death and poor prognosis. It is important that the cardiovascular field be cognizant of the potential effects of modulating B-cell activity on AMI patients. The level of BAFF may be one of the important prognostic indicators for STEMI.

There are several limitations to the present study. Firstly, this was an observational cohort study. Although we used a rational adjustment model consisting of known post-STEMI risk factors, we could not exclude residual confounding. Moreover, the results need to be validated in larger cohort studies with different environmental and genetic backgrounds to further determine the predictive and prognostic value of BAFF in those patients with AMI. Serial evaluations of BAFF levels at different time points in the post-MI period might be complementarily used in the future to assess the specificity and efficiency of BAFF as a diagnostic or prognostic biomarker. In addition, our present research did not prove causality between BAFF and cardiac IR injury. More studies, especially basic research studies, need to be performed to evaluate the role of BAFF in AMI.

## 5. Conclusions

In the present cohort, BAFF elevation showed a markedly adverse prognosis, including cardiovascular death in the post-MI period. The measurement of BAFF during the acute phase could be considered a part of contemporary approaches for risk stratification and may be used as a novel biomarker and predictor of a poor prognosis.

## Figures and Tables

**Figure 1 jcm-12-01692-f001:**
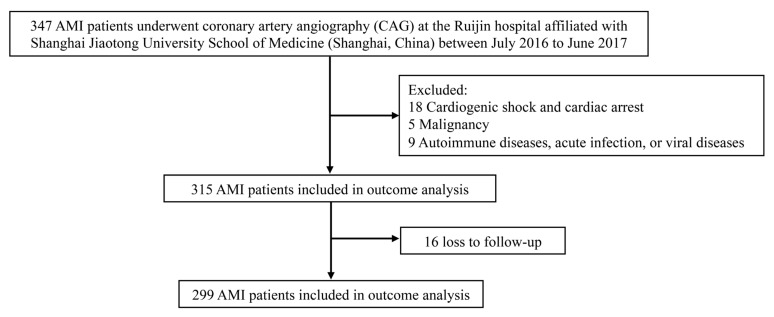
Study Flowchart. Abbreviation: AMI: acute myocardial infarctions.

**Figure 2 jcm-12-01692-f002:**
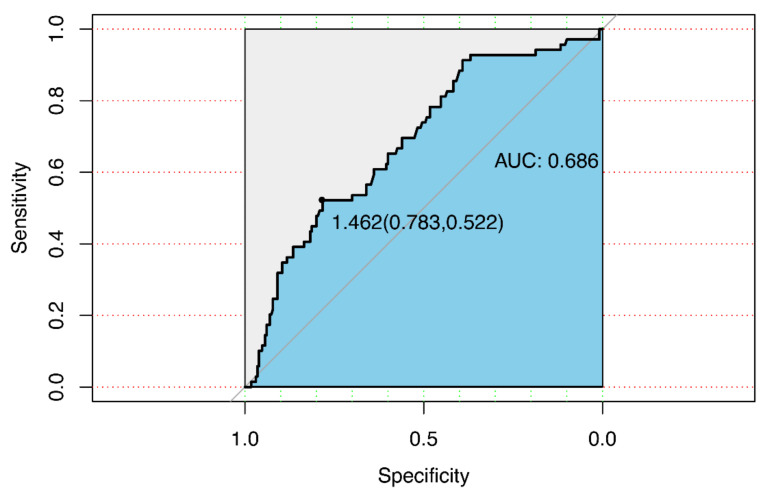
Receiver operating characteristic curve for BAFF in major adverse cardiovascular events. Abbreviation: BAFF, B-cell activating factor; AUC, Area under the curve.

**Figure 3 jcm-12-01692-f003:**
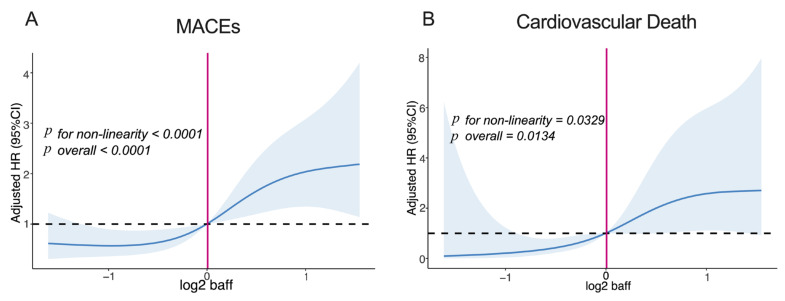
Dose-response relationship between BAFF levels and risk of the composite endpoint. (**A**) Dose-response relationship between BAFF levels and risk of MACEs; (**B**) Dose-response relationship between BAFF levels and risk of cardiovascular death. Abbreviation: Restricted cubic spline curve was carried out with 4 knots at the 5th, 25th, 75th, and 95th percentiles of baseline BAFF levels. The reference point was the median of the BAFF in the 299 participants. The solid line represented point estimation on the association of BAFF with events, and the shaded portion represented 95% CI estimation. Covariates in the model included age, sex, BMI, smoking, history of hypertension, history of DM, history of dyslipidemia, WBC, hs-CRP and LVEF. MACEs: major adverse cardiovascular events; BAFF: B-cell activating factor; BMI: body mass index; hs-CRP: high sensitivity C-reactive protein; WBC: white blood cell; hs-CRP: high sensitivity C-reactive protein; LVEF: left ventricular ejection fraction; DM: diabetes mellitus; HR: hazard ratio; CI: confidence interval.

**Figure 4 jcm-12-01692-f004:**
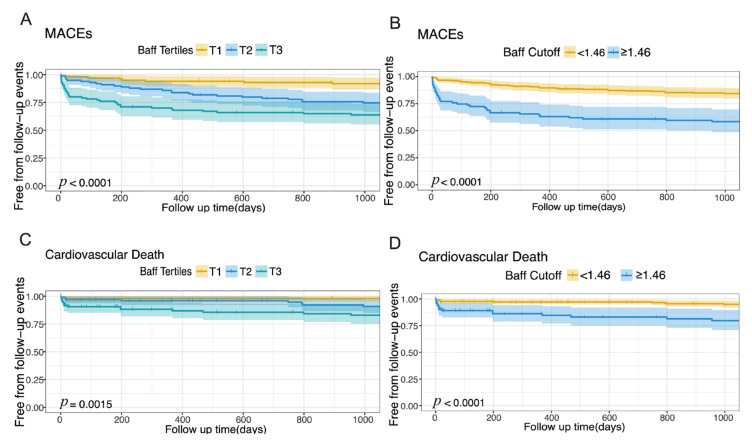
Kaplan–Meier curves of free from follow-up events stratifying by tertiles and a cut-off value of the BAFF levels. (**A**) Kaplan–Meier curve of free from follow-up MACEs stratifying by tertiles of the BAFF levels; (**B**) Kaplan–Meier curve of free from follow-up MACEs stratifying by the cut-off value of the BAFF levels; (**C**) Kaplan–Meier curve of free from follow-up cardiovascular death stratifying by tertiles of the BAFF levels; (**D**) Kaplan–Meier curve of free from follow-up cardiovascular death stratifying by the cut-off value of the BAFF levels. Abbreviation: MACEs: major adverse cardiovascular events; BAFF: B-cell activating factor. Abbreviation: BAFF, B-cell activating factor.

**Figure 5 jcm-12-01692-f005:**
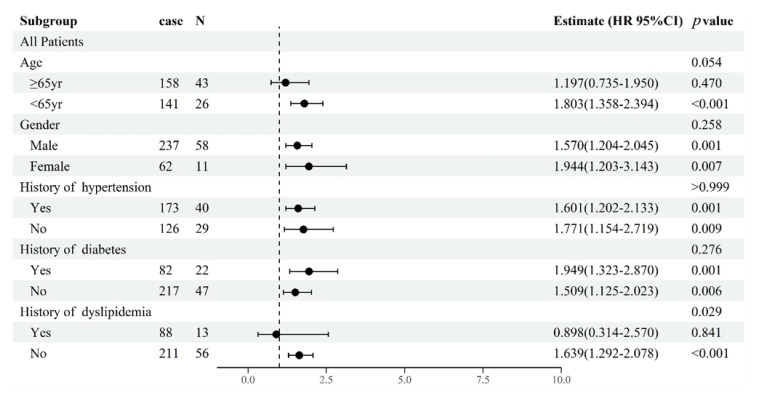
Subgroup analysis to examine the association between the BAFF levels and risk of the composite endpoint in different subgroups of patients. HRs were calculated by multivariable Cox regression analysis adjusting for age, sex, BMI, smoking, history of hypertension, history of DM, history of dyslipidemia, WBC, hs-CRP, and LVEF. Abbreviations: MACEs: major adverse cardiovascular events; BAFF: B-cell activating factor; BMI: body mass index; hs-CRP: high sensitivity C-reactive protein; WBC: white blood cell; hs-CRP: high sensitivity C-reactive protein; LVEF: left ventricular ejection fraction; DM: diabetes mellitus; HR: hazard ratio; CI: confidence interval.

**Table 1 jcm-12-01692-t001:** Baseline characteristics of patients with STEMI by tertiles of BAFF.

	Total (*n* = 299)	BAFF ≤ 0.79 ng/mL (*n* = 100)	0.79 < BAFF ≤ 1.38 ng/mL (*n* = 99)	BAFF > 1.38 ng/mL (*n* = 100)	*p*-Value
**Demographic and risk factors**					
Age (years)	65.43 ± 12.57	59.46 ± 12.02	65.39 ± 12.35	71.44 ± 10.38	<0.001
Male, sex (*n*, %)	237 (79.3)	83 (83.0)	75 (75.8)	79 (79.0)	0.451
BMI (kg/m^2^)	24.08 ± 2.98	24.87 ± 2.60	24.19 ± 2.92	23.18 ± 3.18	<0.001
Smoking (*n*, %)	135 (45.2)	52 (52.0)	47 (47.5)	36 (36.0)	0.064
Alcohol (*n*, %)	75 (25.1)	21 (21.0)	31 (31.3)	23 (23.0)	0.206
Heart rate (beats/minute)	83.21 ± 16.18	80.64 ± 13.35	82.31 ± 15.64	86.48 ± 18.71	0.033
Systolic pressure (mmHg)	123.91 ± 21.46	126.54 ± 20.16	123.65 ± 21.30	121.19 ± 22.89	0.212
Diastolic pressure (mmHg)	74.92 ± 13.60	75.92 ± 14.86	75.22 ± 11.00	73.63 ± 14.60	0.476
**Medical history**					
Hypertension (*n*, %)	173 (57.9)	47 (47.0)	60 (60.6)	66 (66.0)	0.020
Diabetes (*n*, %)	82 (27.4)	26 (26.0)	28 (28.3)	28 (28.0)	0.925
Dyslipidemia (*n*, %)	88 (29.4)	33 (33.0)	30 (30.0)	25 (25.0)	0.450
**Clinical presentation**					
WBC (×109/L)	9.20 (7.10–11.20)	9.50 (7.33–11.90)	9.10 (6.86–10.79)	8.90 (6.98–11.21)	0.454
Hemoglobin (g/L)	137.00 (122.00–147.00)	140.08 (132.00–149.00)	135.00 (122.00–147.00)	130.00 (112.75–142.25)	<0.001
Platelet (×109/L)	190.00 (157.50–221.00)	192.50 (150.00–224.00)	196.00 (170.00–232.00)	180.50 (152.75–210.00)	0.090
HbA1c (%)	6.00 (5.65–6.80)	5.90 (5.50–6.90)	6.20 (5.70–6.78)	6.05 (5.70–6.80)	0.386
Fasting glucose (mmol/L)	6.09 (5.24–7.33)	5.64 (4.98–7.28)	6.09 (5.25–7.50)	6.45 (5.58–7.50)	0.037
hs-CRP (mg/L)	3.00 (1.07–7.96)	2.50 (1.11–5.08)	2.80 (0.72–6.00)	6.16 (1.30–19.21)	<0.001
Triglyceride (mmol/L)	1.29 (0.97–1.82)	1.42 (1.083–2.09)	1.33 (1.02–1.74)	1.22 (0.87–1.60)	0.100
Total cholesterol (mmol/L)	4.58 (3.72–5.36)	4.49 (3.86–5.39)	4.73 (3.73–5.59)	4.50 (3.36–5.27)	0.107
HDL-C (mmol/L)	1.05 (0.92–1.25)	1.08 (0.90–1.21)	1.03 (0.94–1.30)	1.04 (0.92–1.25)	0.962
LDL-C (mmol/L)	2.87 (2.22–3.52)	3.03 (2.47–3.46)	3.01 (2.32–3.70)	2.78 (2.05–3.26)	0.091
Lp(a) (mmol/L)	0.14 (0.07–0.29)	0.11 (0.07–0.27)	0.15 (0.08–0.25)	0.14 (0.06–0.38)	0.440
Creatine (μmol/L)	81.00 (70.00–100.00)	77.00 (68.25–88.00)	76.00 (67.00–76.00)	97.50 (77.00–136.00)	<0.001
Cystatin C (mg/L)	1.04 (0.91–1.29)	0.97 (0.88–1.11)	1.01 (0.91–1.24)	1.21 (0.97–1.57)	<0.001
eGFR (mL/minute/1.73 m^2^)	82.75 (62.43–94.80)	91.05 (79.68–98.93)	87.85 (72.85–95.95)	61.95 (43.38–81.88)	<0.001
NT-proBNP (pg/mL)	614.95 (152.85–2026.00)	406.00 (130.00–1118.00)	292.10 (107.80–1282.00)	1551.50 (392.30–4637.75)	<0.001
CK-MB (mg/L)	107.60 (26.30–275.00)	103.65 (15.18–211.35)	66.50 (22.70–266.10)	194.70 (60.23–318.40)	0.001
cTnI (ng/L)	20.27 (6.06–61.22)	17.14 (3.59–49.71)	15.80 (2.83–55.93)	36.76 (11.71–82.00)	<0.001
Syntax score	19.00 (12.00–27.00)	17.00 (8.00–26.13)	19.00 (13.00–26.50)	21.00 (15.50–29.00)	0.003
**Echocardiography**					
LVEF (%)	56.47 ± 8.20	58.08 ± 6.82	58.09 ± 7.09	53.11 ± 9.54	<0.001
LAD (mm)	38.08 ± 3.91	37.44 ± 3.20	37.80 ± 3.66	39.03 ± 4.64	0.012
LVEDD (mm)	49.31 ± 4.16	49.06 ± 3.57	48.94 ± 3.94	49.96 ± 4.86	0.182
LVESD (mm)	34.13 ± 4.69	33.23 ± 3.39	33.53 ± 4.56	35.70 ± 5.56	<0.001
LVEDV (mL)	116.80 ± 23.58	115.05 ± 19.00	114.99 ± 22.59	120.50 ± 28.26	0.180
LVESV (mL)	50.17 ± 19.45	46.63 ± 11.33	47.81 ± 14.84	56.31 ± 22.92	<0.001
**Ischemia time**					0.170
0 h ≤ chest pain to balloon ≤ 6 h	100 (33.4)	41 (41.0)	32 (32.3)	27 (27.0)	
6 h < chest pain to balloon ≤ 12 h	99 (33.1)	34 (34.0)	31 (31.3)	34 (34.0)	
12 h < chest pain to balloon ≤ 24 h	100 (33.4)	36 (36.0)	30 (30.3)	34 (34.0)	
**Culprit vessel**					0.013
LAD (*n*, %)	163 (54.5)	60 (60.6)	55 (55.0)	48 (48.0)	
LCX (*n*, %)	32 (10.7)	8 (8.1)	15 (15.0)	9 (9.0)	
RCA (*n*, %)	92 (30.7)	25 (25.3)	30 (30.0)	37 (37.0)	
**Number of diseased vessels**					0.391
Single vessel	63 (21.4)	23 (23.0)	32 (32.0)	44 (44.0)	
Double vessel	81 (27.6)	24 (24.0)	25 (25.3)	50 (50.5)	
Triple vessel	149 (50.7)	16 (16.0)	24 (25.3)	55 (57.9)	
**Killip classification**					<0.001
I (*n*, %)	205 (68.6)	89 (89.0)	81 (81.8)	35 (35.0)	
II (*n*, %)	57 (19.1)	8 (8.0)	13 (13.1)	36 (36.0)	
III/IV (*n*, %)	37 (12.4)	3 (3.0)	5 (5.1)	29 (29.0)	
**Medication**					
Aspirin (*n*, %)	289 (96.7)	99 (99.0)	96 (97.0)	94 (94.0)	>0.999
Clopidogrel (*n*, %)	251 (86.9)	79 (79.0)	91 (91.9)	81 (81.0)	0.008
Ticagrelor (*n*, %)	39 (13.6)	20 (20.0)	5 (5.1)	14 (14.0)	0.010
Statins (*n*, %)	283 (97.9)	97 (97.0)	96 (97.0)	90 (90.0)	0.120
Anticoagulants (*n*, %)	109 (37.0)	25 (25.0)	41 (41.4)	41 (41.0)	0.011
ACEI/ARB (*n*, %)	244 (84.4)	81 (81.0)	90 (90.9)	73 (73.0)	0.006
Beta blockers (*n*, %)	268 (92.7)	93 (93.0)	91 (91.9)	84 (84.0)	0.301
Nitrates (*n*, %)	106 (36.6)	33 (33.0)	34 (34.3)	39 (39.0)	0.515

Values are mean ± SD, *n* (%), or median (first quartile, third quartile). T1, BAFF ≤ 0.79 ng/mL; T2, 0.79 ng/mL < BAFF ≤ 1.38 ng/mL; T3, BAFF > 1.38 ng/mL; ACEI: angiotensin-converting enzyme inhibitor; ARB: angiotensin receptor blocker; BAFF: B-cell activating factor; BMI: body mass index; CK-MB: creatine kinase-MB isoenzyme; cTnl” Cardiac troponin I; eGFR: estimated glomerular filtration; HbA1c: glycated hemoglobin; HDL-C: high-density lipoprotein cholesterol; hs-CRP, high sensitivity C-reactive protein; LAD: left atrial diameter; LDL-C: low-density lipoprotein cholesterol; LVEDD: left ventricular end-diastolic diameter; LVEDV: left ventricular end-diastolic volume; LVEF: left ventricular ejection fraction; LVESD: left ventricular end-systolic diameter; LVESV: left ventricular end-systolic volume; NT-proBNP: N-terminal pro-brain natriuretic peptide; WBC: white blood cell.

**Table 2 jcm-12-01692-t002:** Cox hazard models for BAFF with MACEs in patients with STEMI.

	Unadjusted HR (95%CI)	*p*-Value	Adjusted for Model 1 HR (95%CI)	*p*-Value	Adjusted for Model 2 HR (95%CI)	*p*-Value
**MACEs**						
**Log_2_ BAFF**	1.828 (1.383–2.415)	<0.001	1.628 (1.215–2.183)	0.001	1.525 (1.085–2.145)	0.015
**BAFF cut-off**	3.439 (2.143–5.521)	<0.001	2.864 (1.747–4.694)	<0.001	2.705 (1.518–4.822)	0.001
**Tertiles of BAFF**						
T1	1 (Ref)		1 (Ref)		1 (Ref)	
T2	3.380 (1.525–7.495)	0.003	3.061 (1.370–6.839)	0.006	3.341 (1.412–7.903)	0.006
T3	5.602 (2.603–12.058)	<0.001	4.373 (1.973–9.691)	<0.001	4.116 (1.690–10.022)	0.002
**Cardiovascular death**						
**Log_2_ BAFF**	2.366 (1.482–3.779)	<0.001	1.957 (1.181–3.242)	0.009	2.083 (1.014–4.279)	0.046
**BAFF cut-off**	4.594 (2.059–10.245)	<0.001	3.102 (1.351–7.120)	0.008	3.632 (1.132–11.650)	0.030
**Tertiles of BAFF**						
T1	1 (Ref)		1 (Ref)		1 (Ref)	
T2	4.336 (0.920–20.425)	0.064	3.329 (0.701–15.802)	0.130	4.330 (0.491–38.208)	0.187
T3	9.112 (2.082–39.886)	0.003	5.107 (0.944–8.767)	0.063	6.607 (0.755–57.830)	0.088

BAFF was transformed into logarithmic form, a categorical variable using the lowest tertile as the reference. MACE indicates a composite endpoint of cardiovascular death, heart failure re-hospitalization, recurrent myocardial infarction, or nonfatal stroke. Model 1 was adjusted for age and gender; Model 2 was adjusted for model 1 and BMI, smoking, history of hypertension, history of DM, history of dyslipidemia, WBC, hsCRP and LVEF. BAFF: B-cell activating factor; BMI: body mass index; hs-CRP: high sensitivity C-reactive protein; WBC: white blood cell; hs-CRP, high sensitivity C-reactive protein; LVEF: left ventricular ejection fraction; DM: diabetes mellitus; HR: hazard ratio; and CI: confidence interval.

**Table 3 jcm-12-01692-t003:** Accuracy of risk prediction of MACEs using cTnI, NT-proBNP, CK-MB and BAFF.

	C-Statistic	*p*-Value	IDI (95%CI)	*p*-Value
Log_2_ cTnI	0.623	NA	Ref	NA
Log_2_ BAFF	0.680	0.263	0.030 (0.001, 0.058)	0.044
Log_2_ NT-proBNP	0.612	0.807	0.007 (−0.015, 0.029)	0.536
Log_2_ CK-MB	0.588	0.114	−0.013 (−0.023, −0.004)	0.008
Log_2_ cTnI+ Log_2_ BAFF	0.675	0.126	0.042 (0.020, 0.064)	<0.001
Log_2_ cTnI+ Log_2_ NTproBNP+ Log_2_ CK-MB+ Log_2_ BAFF	0.675	0.158	0.059 (0.031, 0.086)	<0.001

BAFF, NT-proBNP, CK-MB, and cTnI were transformed into logarithmic form.

## Data Availability

The raw data supporting the conclusions of this article will be made available by the authors without undue reservation.

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
