# Peer review of "Association of Serum BAFF Levels with Cardiovascular Events in ST-Segment Elevation Myocardial Infarction"

_jcm, 2023, doi:10.3390/jcm12041692_

Round 1
Reviewer 1 Report
In the present study, the authors have identified BAFF as a potential prognostic marker for determining the clinical outcome of STEMI patients. Using a cohort of 299 patients, the authors have established BAFF as an independent pedictor of the of MACE in AMI patients. There are a few minor revisions which may improve the clarity and therefore impact of the study:
1. A stronger rationale for studying BAFF as a prognostic marker needs to be provided since no prior studies exist.
2. How was the cut off value for BAFF determined in the study?
3. Font size in the Figures in supplementary files must be increased to make it easily readable.
Reviewer 2 Report
The paper is interesting, and the topic of modern markers and predictors always raises many questions.
The authors presented an interesting study, but the reviewer has a few comments.
1. The addition of a flowchart would have improved the depiction of the follow-up period.
2. The analysis of the effect of BAFF concentration on the occurrence of study endpoints does not provide complete information about the absolute effect of all relevant factors so that the reader can make a comparison.
3.The authors described the limitations of this study what was placed in the discussion. Looking at the coincidence of high BAFF levels with the severity of known factors of poor prognosis, it is difficult to conclude that the value of this marker is unique. It seems that this marker could be more helpful in the selection of patients with chronic coronary syndrome with the highest risk of MACEs and death, where known poor prognosis factors do not manifest themselves to a large extent resulting from STEMI.
4. The authors did not provide baseline count times in STEMI, e.g., pain to balloon, pain to FMC. In addition, despite the fact that more than 50% of cases were LAD related the LVEF was surprisingly good 2 days after admission, this would suggest that the ischemic time was very short. The reviewer asks that this issue be addressed.
5. The authors should consider adding to the discussion articles relating the strongest predictors of long-term survival in STEMI, for example:
Kaziród-Wolski K, Sielski J, GÄ…sior M, et al. Factors affecting short- and long-term survival of patients with acute coronary syndrome treated invasively using intravascular ultrasound and fractional flow reserve: Analysis of data from the Polish Registry of Acute Coronary Syndromes 2017-2020 [published online ahead of print, 2022 Nov 21]. Kardiol Pol. 2022;10.33963/KP.a2022.0261. doi:10.33963/KP.a2022.0261
Köktürk U, PüÅŸüroÄŸlu H, Somuncu MU, et al. Short and Long-Term Prognostic Significance of Galectin-3 in Patients with ST-Elevation Myocardial Infarction Undergoing Primary Percutaneous Coronary Intervention [published online ahead of print, 2023 Jan 3]. Angiology. 2023;33197221149846. doi:10.1177/00033197221149846
